# Fractional Langevin Dynamics for Combinatorial Optimization via Polynomial-Time Escape

**Shiyue Wang**[1,3†]**, Ziao Guo**[2†]**, Changhong Lu**[1]**, Junchi Yan**[2,3] *

[1]School of Mathematical Sciences, Key Laboratory of MEA,
and Shanghai Key Laboratory of PMMP, East China Normal University
[2]School of Computer Science & School of Artificial Intelligence, Shanghai Jiao Tong University
[3]Shanghai Innovation Institute
{wangshiyue@stu,chlu@math}.ecnu.edu.cn  {ziao.guo,yanjunchi}@sjtu.edu.cn

## Abstract

Langevin dynamics (LD) and its discrete proposal have been widely applied in the field of Combinatorial Optimization (CO). Both sampling-based and data-driven approaches have benefited significantly from these methods. However, LD's reliance on Gaussian noise limits its ability to escape narrow local optima, requires costly parallel chains, and performs poorly in rugged landscapes or with non-strict constraints. These challenges have impeded the development of more advanced approaches. To address these issues, we introduce fractional Langevin dynamics (FLD) for CO, replacing Gaussian noise with $\alpha$-stable Lévy noise. FLD can escape from local optima more readily via Lévy flights, and in multiple-peak CO problems with high potential barriers it exhibits a polynomial escape time that outperforms the exponential escape time of LD. Moreover, FLD coincides with LD when $\alpha = 2$, and by tuning $\alpha$ it can be adapted to a wider range of complex scenarios in the CO field. We provide theoretical proof that our method offers enhanced exploration capabilities and improved convergence. Experimental results on the Maximum Independent Set, Maximum Clique, and Maximum Cut problems demonstrate that incorporating FLD advances both sampling-based and data-driven approaches, achieving state-of-the-art (SOTA) performance in most of the experiments. The codes are publicly available at https://github.com/Thinklab-SJTU/FLD4CO.

## 1 Introduction

Combinatorial optimization (CO) problems, which involve finding an optimal solution from a finite set of possible configurations subject to a set of constraints, are of paramount importance and usefulness across fields, e.g. logistics [45], scheduling [59], network design [4], and finance [39].

There has been growing interest in developing efficient algorithms for obtaining high-quality sub-optimal solutions. Among these efforts, sampling-based methods have shown considerable promise due to their simplicity, ability to balance speed and solution quality, and training-free property. A fundamental approach is simulated annealing (SA) [28], which uses random local fluctuations guided by Metropolis-Hastings updates [36, 23] and probabilistically explores the solution space. Recent work [51] has demonstrated that incorporating Langevin dynamics (LD) and its discrete proposal [69, 50] can vastly improve the sampling efficiency, thereby advancing sampling-based approaches for CO. The core idea of LD is to leverage the gradient to guide the sampling in each iteration, resulting in a more efficient searching process. However, there are certain limitations

---

*Correspondence author. † denotes equal contribution. This work was partly supported by National Key R&D Program of China (Nos. 2021YFA1000300 and 2021YFA 100302) and National Natural Science Foundation of China (No. 12331014, 62222607).

associated with LD. Firstly, it relies on Gaussian noise as a random perturbation, with the step size being coupled to the noise amplitude. As the Gaussian noise decays exponentially at the tail, reducing the step size also diminishes the noise, making it challenging to escape from 'deep and narrow' local optima. The time required to escape local minima grows exponentially with the energy barrier height. Moreover, to maintain sample diversity, LD necessitates parallel independent chains, which can be computationally expensive. Furthermore, Gaussian noise assumes a locally smooth energy landscape, meaning that LD is less effective in scenarios where the energy function is rugged or when non-strict constraints are present. These factors limit the effectiveness of LD in more complex optimization landscapes and the development of more advanced sampling-based approaches for CO.

Another trend is the data-driven approach to learning for optimization. Early neural network (NN)-based methods [24] primarily relied on supervised learning [33, 18, 21]. Subsequent works have explored reinforcement learning [62, 60, 61] and unsupervised learning [27, 57, 52] techniques to address the challenge of collecting labeled training data. More recently, diffusion models have been introduced to the CO domain [31, 53, 44, 43], demonstrating superior performance and promising potential. These approaches also implicitly incorporate LD, as diffusion models were initially derived from LD in the field of image generation. Unlike sampling-based methods, NN-based approaches eliminate the need for explicit gradients of the problem, thereby enabling unification for a variety of problems without relying on the problem structure, utilizing the network's automatic differentiation capabilities. We leave detailed related works in Appendix 2.

In this paper, we introduce fractional Langevin dynamics (FLD) to address the propensity of conventional LD to become trapped in local optima. We incorporate symmetric $\alpha$-stable ($\mathcal{S}\alpha\mathcal{S}$) noise with truncation into FLD: unlike Gaussian perturbations, $\mathcal{S}\alpha\mathcal{S}$ noise exhibits heavy-tailed jumps (Lévy flights), enabling instantaneous energy-barrier leaps that facilitate escape from local minima. Moreover, by setting $\alpha = 2$, FLD reduces to standard LD, thus retaining efficient exploration in smoother or strongly constrained settings. We propose the $\mathcal{S}\alpha\mathcal{S}$-noise FLD sampling process and present both explicit- and implicit-gradient formulations to advance both sampling-based and data-driven approaches. We adopt the mean escape time as our convergence metric, and derive theoretical upper bounds in the discrete setting, showing a polynomial-time bound for FLD versus an exponential bound for LD. Through comparative case studies on three prototypical CO problems, our methods outperform existing sampling-based and data-driven methods. Additionally, extensive sampling-trajectory experiments have been conducted to vividly illustrate the enhanced escape ability of FLD, demonstrate the impact of varying $\alpha$ on escape performance, and confirm the effectiveness of our truncation strategy. Finally, we perform ablation studies on the best energy-function values over iterations, thereby validating superior convergence and exploration capabilities of FLD.

## 2 Related Work

**Data-driven Approaches for CO.** They involve training NN models for CO, commonly referred to as neural solvers. Significant efforts have been made to explore supervised learning [33, 18, 21, 53, 31, 58, 32, 30, 34, 35], unsupervised learning [27, 57, 52, 56, 44, 43, 20], and reinforcement learning [40, 17]. Our FLD-IG integrates FLD with a simple reinforcement learning-based approach. FLD-IG achieves competitive performance with a simple architecture and minimal training resources, resulting in faster convergence and improved training efficiency.

**Sampling-based Approaches for CO.** Sampling-based approaches have been widely utilized for CO [36, 23, 37, 10, 65, 46]. However, these previous approaches are generally less efficient than data-driven approaches. Recent work by [51] has advanced sampling-based methods, achieving comparable or even superior performance to data-driven approaches by introducing the discrete LD proposal[69, 50]. [17] further develops a regularized approach on discrete LD, resulting in improved performance. Our FLD-EG enhances the sampling-based approach by integrating FLD, which can be seen as a generalization of vanilla LD, leading to faster convergence and better performance.

## 3 Preliminaries

**Energy-Based Model (EBM).** It defines an energy function $H : \mathcal{S} \to \mathbb{R}$ with the target distribution:

$$p_\tau(x) : \frac{e^{-H(x)/\tau}}{Z} \tag{1}$$

where $\mathcal{S}$ represents the energy state space, $\tau$ is a temperature parameter controlling the smoothness of the distribution, and $Z = \sum_{x \in \mathcal{S}} e^{-H(x)/\tau}$ is the partition function in statistical physics or normalization factor in probability theory.

**Markov Chain Monte Carlo.** Markov chain Monte Carlo (MCMC) techniques [63, 54], which are grounded in continuous diffusion processes, have gained widespread popularity owing to their demonstrated success in large-scale Bayesian machine learning [11]. The goal of the MCMC is to generate samples from a target distribution $p_\tau$, by forming a continuous diffusion that has $p_\tau$ as a stationary distribution. Given a current state $x_t \in \mathcal{S}$, a Metropolis-Hastings (MH) sampler [36, 23] proposes a candidate state $y \in \mathcal{S}$ from a proposal distribution $g(y \mid x_t)$. Then calculate the Metropolis acceptance ratio:

$$A(y, x_t) = \min \left\{ 1, \frac{p_\tau(y)g(x_t \mid y)}{p_\tau(x_t)g(y \mid x_t)} \right\} \tag{2}$$

With generating a random number $u \sim U(0,1)$, where $U(0,1)$ is the uniform distribution within $[0, 1]$, if $u \leq A(y, x_t)$, then the proposal state is accepted and set $x_{t+1} = y$; otherwise, set $x_{t+1} = x_t$.

**Langevin Dynamics.** Langevin dynamics (LD) is an MCMC algorithm that has also been incorporated in combinatorial optimization algorithms for better exploring the landscape of the energy function $H(x)$ [47]. LD methods are based on constructing stochastic differential equations (SDEs) equipped with Brownian motion (shown as Eq. (3)), assuming that the particle is driven by an infinite number of small forces with finite variance.

$$dx_t = s(x_t)dt + \sqrt{2}dB_t \tag{3}$$

where $B_t$ denotes the standard Brownian motion and $s(\cdot) = \nabla \log p_\tau(\cdot) = -\frac{1}{\tau}\nabla H(x)$ represents the score function of EBM. With the condition of sampling state $x_t$ can be shown to be ergodic with $p_\tau(x_t)$. The samples can be generated from $p_\tau$ by simulating the sampling process of continuous space discrete space [41], which is given by using a first-order Euler-Maruyama discretization:

$$x_{n+1} = x_n + \eta_{n+1}s(x_n) + \sqrt{2\eta_{n+1}}\Delta B_{n+1} \tag{4}$$

where $\eta_n$ denotes the step size of the sampling iteration and $\Delta B_n = \xi$ is an i.i.d. standard Gaussian random variable, $\xi \sim \mathcal{N}(0, I_{N \times N})$ when the state space $\mathcal{S} = \mathbb{R}^N$ [69].

**Simulated Annealing.** Simulated annealing (SA) is a variant of local search [14] that explores the solution landscape with probabilistic relaxation. As the temperature decreases, there is a tendency to sample points on the landscape to make the energy function $H(x)$ value smaller; when the temperature equals zero, the solution $x$ will stop at the point where the $H(x)$ has the lowest value (that is, the solution obtained by the SA algorithm converges to the global optimum in probability) [55].

## 4 Methodology

### 4.1 Problem Formulation

Without loss of generality, we formulate a CO problem as follows:

$$\min_{x \in \mathcal{S} = \{0,1\}^N} a(x), \text{ s.t. } b(x) = 0 \tag{5}$$

where the solution landscape $\mathcal{S}$ is an $N$-dimensional vector such that each dimension takes a discrete value from $\{0, 1\}$, which is the most challenging to deal with, although it will be possible to extend.

To recast a constrained optimization problem as a sampling task, a penalty function (generally treated as the energy function of EBM) takes the form:

$$H(x) = a(x) + \lambda b(x) \tag{6}$$

where $\lambda$ is the penalty factor of the constraints. Furthermore, the attempt to directly sample from $p_\tau(x)$ with the small $\tau$ makes the energy landscape highly nonsmooth; a common remedy is to incorporate the SA algorithm, progressively lowering $\tau$ toward zero as the chain evolves.

## 4.2 Fractional Langevin Dynamics

By Eq. (4), it can be seen that the term $(x_{n+1} - x_n - \eta_{n+1}s(x_n))/\sqrt{2\eta_{n+1}}$ follows a Gaussian. Thus the transition probability $q(x_{n+1} \mid x_n)$ in the LD algorithm can be interpreted as a Gaussian with mean $x_n + \eta_{n+1}s(x_n)$ and covariance $2\eta_{n+1}I_{N \times N}$ [69]. The discrete (gradient-based) proposal distribution with the explicit domain $\mathcal{S} = \mathbb{R}^N$ of LD is:

$$q(x_{n+1} \mid x_n) = \frac{\exp\left(-\frac{1}{4\eta_{n+1}}\|x_{n+1} - x_n - \eta_{n+1}s(x_n)\|_2^2\right)}{Z_{\mathbb{R}^N}(x_n)} \tag{7}$$

where,

$$Z_{\mathbb{R}^N}(x_n) = \sum_{x_{n+1}\in\mathbb{R}^N} \exp\left(-\frac{1}{4\eta_{n+1}}\|x_{n+1} - x_n - \eta_{n+1}s(x_n)\|_2^2\right) = (4\pi\eta_{n+1})^{N/2} \tag{8}$$

Thus, it can be factorized coordinate-wise into a set of simple categorical distributions:

$$q(x_{i,n+1} \mid x_n) \propto \exp\left\{\frac{1}{2}s(x)_i(x_{i,n+1} - x_{i,n} - \frac{(x_{i,n+1} - x_{i,n})^2}{\eta_{n+1}})\right\} \tag{9}$$

with

$$q(x_{n+1} \mid x_n) = \prod_{i=1}^{N} q(x_{i,n+1} \mid x_n) \tag{10}$$

Typically, the iteration step size $\eta_n = C$, where $C$ is a constant, is too stable due to the combination of a fixed step size and Gaussian noise. This stability makes it difficult for LD to escape from local optima on the energy surface during the iterative process, often causing the trajectory to remain trapped near suboptimal solutions, thereby significantly degrading the quality of the final result. To address this issue, we introduce the $\alpha$-stable Lévy noise [67]—a type of stochastic process with a heavy-tailed distribution. Unlike Gaussian noise, $\alpha$-stable Lévy noise allows for occasional large jumps (Lévy flights), which increases the probability of escaping local optima and thus improves the exploration capability of the algorithm.

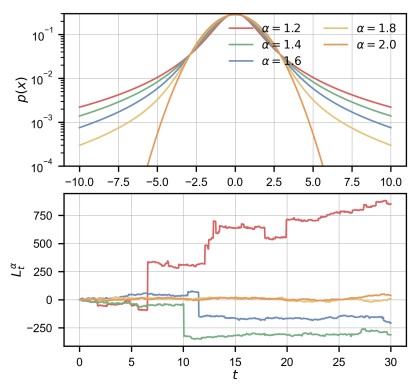

Figure 1: Pdf of $\mathcal{S}\alpha\mathcal{S}$ distribution and $\alpha$-stable Lévy motion.

In this work, we are interested in the centered symmetric $\alpha$-stable ($\mathcal{S}\alpha\mathcal{S}$) distribution, which is a special case of $\alpha$-stable distribution. The definition of $\mathcal{S}\alpha\mathcal{S}$ random variables and $\mathcal{S}\alpha\mathcal{S}$ Lévy motion are shown as:

**Definition 1** ($\mathcal{S}\alpha\mathcal{S}$ random variables [42]). *The $\alpha$-stable distribution arises as the limiting distribution in the generalized Central Limit Theorem (CLT). A scalar random variable $x \in \mathbb{R}$ is said to follow a $\mathcal{S}\alpha\mathcal{S}$ distribution if its characteristic function takes the following form:*

$$\mathbb{E}[\exp(iwx)] = \exp(-|\sigma w|^\alpha) \tag{11}$$

*Here, $\alpha \in (0, 2]$ is the characteristic exponent, which controls the tail heaviness of the distribution: smaller values of $\alpha$ result in heavier tails (shown in Figure 1). The parameter $\sigma \in \mathbb{R}^+$ is the scale parameter, reflecting the dispersion of $x$ around zero.*

**Definition 2** ($\mathcal{S}\alpha\mathcal{S}$ Lévy motion [15]). *A scalar symmetric $\alpha$-stable Lévy motion $L_t^\alpha$, with $\alpha \in (0, 2]$, is a stochastic process satisfying the following properties:*

1. *$L_0^\alpha = 0$ almost surely.*

2. *For $t_0 < t_1 < \cdots < t_N$, the increments $(L_{t_n}^\alpha - L_{t_{n-1}}^\alpha)$ $(n = 1, 2, \cdots, N)$ are independent.*

3. *The $(L_t^\alpha - L_s^\alpha)$ and $L_{t-s}^\alpha$ have the same distribution $\mathcal{S}\alpha\mathcal{S}((t - s)^{1/\alpha})$ $(0 \leq s < t)$.*

4. $L_t^\alpha$ has stochastically continuous sample paths (that is, continuous in probability):

$$\lim_{t \to s} \mathbb{P}(|L_t^\alpha - L_s^\alpha| > \delta) = 0, \ \forall \delta > 0, s \geq 0. \tag{12}$$

Similarly to the $\mathcal{S}\alpha\mathcal{S}$ distributions, the $\mathcal{S}\alpha\mathcal{S}$ Lévy motions $L_t^\alpha$ coincide with a scaled Brownian motion $\sqrt{2}B_t$ when $\alpha = 2$. Thus, the $\mathcal{S}\alpha\mathcal{S}$ distribution can be seen as a heavy-tailed generalization of the centered Gaussian distribution. As an important special case of $\mathcal{S}\alpha\mathcal{S}$, we obtain the Gaussian distribution $\mathcal{S}\alpha\mathcal{S}(\sigma) = \mathcal{N}(0, 2\sigma^2)$ for $\alpha = 2$.

The fractional Langevin dynamics (FLD) framework is driven by the $\mathcal{S}\alpha\mathcal{S}$ Lévy-based SDE as [47]:

$$dx_t = b(x_{t-}, \alpha)dt + dL_t^\alpha \tag{13}$$

where, $b(\cdot)$ denotes the drift function shown in Theorem 1, $x_{t-}$ represents the left limit of the process at time $t$, and $L_t^\alpha$ is the standard $\mathcal{S}\alpha\mathcal{S}$ Lévy motion shown as Definition 2.

**Theorem 1** ([47]). *The drift function of the SDE (13) is defined below:*

$$b(x, \alpha) \triangleq \frac{\mathcal{D}^{\alpha-2} f_{p_\tau}(x)}{\phi(x)} \triangleq c_\alpha s(x) \tag{14}$$

*where, $f_{p_\tau}(x) \triangleq -\phi(x)\partial_x H(x)$, fractional integration $\mathcal{D}^{\alpha-2} \triangleq \mathcal{F}^{-1}\{|w|^{\alpha-2}\mathcal{F}(f_{p_\tau}(x))\}$, $\phi(x) = \exp\{-H(x)\}$, $c_\alpha \triangleq \Gamma(\alpha-1)/\Gamma(\alpha/2)^2$, and $\mathcal{F}$ is the Fourier transforms.*

Detailed derivation and proof can be found in Appendix A.

**Proposition 1.** *The FLD-based SDE is the generalization of the LD-based SDE.*

*Proof.* When $\alpha = 2$, the SDE driven by the FLD:

$$dx_t = b(x_{t-}, 2)dt + dL_t^\alpha(\alpha \to 2) \approx c_2 s(x_t)dt + dL_t^\alpha(\alpha \to 2) \approx s(x_t)dt + \sqrt{2}dB_t \tag{15}$$

Thus, the FLD-based SDE reduces to the LD-based SDE when $\alpha = 2$, while for $\alpha \neq 2$, the FLD-based SDE exhibits heavier tails. $\qquad\square$

Combining the Theorem 1 and Eq. (13), the approximate $\mathcal{S}\alpha\mathcal{S}$ Lévy-based SDE and the first-order Euler-Maruyama discretized $\mathcal{S}\alpha\mathcal{S}$ sampling process can be obtained as:

$$dx_t = c_\alpha s(x_t)dt + dL_t^\alpha x_{n+1} = \quad x_n + \eta_{n+1} c_\alpha s(x_n) + \eta_{n+1}^{1/\alpha} \triangle L_{n+1}^\alpha \tag{16}$$

The Proposition 1 also demonstrates that the target distribution can be sampled more accurately by adaptively adjusting $\alpha$ during the sampling process. In regions where the energy function is locally smooth or tightly constrained, setting $\alpha = 2$ enables efficient sampling. Conversely, when the sampling process becomes trapped in a local optimum, decreasing $\alpha$ increases the probability of Lévy flights, thereby facilitating escape from the local minimum.

Moreover, although the probability density function (pdf) of the $\mathcal{S}\alpha\mathcal{S}$ distribution does not have a closed-form expression, it is straightforward to generate random samples from stable distributions when $\alpha \neq 2$. The sampling of $\mathcal{S}\alpha\mathcal{S}$ is given by Theorem 2 with $\beta = 0$ by the Chambers-Mallows-Stuck method, which is shown in Theorem 3.

**Theorem 2.** *Let $\gamma$ be uniformly distributed on $(-\frac{\pi}{2}, \frac{\pi}{2})$ and $W$ be an independent exponential random variable with mean 1. The $\alpha$-stable sampling is:*

$$Z = \begin{cases} \dfrac{\sin \alpha(\gamma - \gamma_0)}{(\cos \gamma)^{1/\alpha}} \left( \dfrac{\cos(\gamma - \alpha(\gamma - \gamma_0))}{W} \right)^{(1-\alpha)/\alpha} \triangleq \mathcal{S}_\alpha(1, \beta, 0), \ \alpha \neq 1 \\[3mm] (\dfrac{\pi}{2} + \beta\gamma) \tan \gamma - \beta \log \left( \dfrac{W \cos \gamma}{\frac{\pi}{2} + \beta\gamma} \right) \triangleq \mathcal{S}_1(1, \beta, 0), \ \alpha = 1 \end{cases} \tag{17}$$

*where, $\gamma_0 = -\frac{\pi \beta K(\alpha)}{2\alpha}$, $K(\alpha) = \alpha - 1 + \text{sign}(1-\alpha)$, and $\text{sign}(\cdot)$ denotes the sign function.*

**Theorem 3.** *Let $\gamma$ be uniformly distributed on $(-\frac{\pi}{2}, \frac{\pi}{2})$ and $W$ be an independent exponential random variable with mean 1. The $\mathcal{S}\alpha\mathcal{S}$ sampling is:*

$$Z = \begin{cases} \dfrac{\sin \alpha\gamma}{(\cos \gamma)^{1/\alpha}} \left( \dfrac{\cos(\gamma - \alpha\gamma)}{W} \right)^{(1-\alpha)/\alpha} \triangleq \mathcal{S}\alpha\mathcal{S}(1), \ \alpha \neq 1 \\[3mm] \dfrac{\pi}{2} \tan \gamma \triangleq \mathcal{S}1\mathcal{S}(1), \ \alpha = 1 \end{cases} \tag{18}$$

Since $\mathcal{S}\alpha\mathcal{S}$ distributions are a special case of $\alpha$-stable distributions, the detailed proof of Theorem 2 and Theorem 3 is presented together in Appendix B. Thus the discrete sampling process can be rewritten as [48], where $z_{n+1} \sim \mathcal{S}\alpha\mathcal{S}(1)$:

$$x_{n+1} = x_n - \frac{\eta_{n+1}c_\alpha}{\tau}\nabla H(x) + \eta_{n+1}^{1/\alpha}z_{n+1} \tag{19}$$

### 4.3 Comparative Analysis of Convergence

We compare the convergence capabilities of LD and FLD by analyzing the escape time from the local minima, which is defined as follows [5]:

**Definition 3** (Escape Time). *The escape time is a random variable:*

$$\tau_{y^*}^{x^*} = \inf\{t > 0 : x_t \in \mathcal{B}_\delta(y^*)\} \tag{20}$$

*where, $x_0 = x^*$, two points $x^*$ and $y^*$ separately represent local minima under the assumption that the potential energy $H$ has several (at least two) local minima, and $\mathcal{B}_\delta(y^*)$ denotes the ball of radius $\delta$ centered in $y^*$.*

Under the low noise intensity $\epsilon$, the LD-based SDE can be rewritten as:

$$dx_t = -\nabla H(x_t)dt + \sqrt{2\epsilon}dB_t \tag{21}$$

By Eyring–Kramers law, the mean escape time of LD-based SDE in the continuous space $\mathcal{S} \in \mathbb{R}^N$ is:

$$\mathbb{E}[\tau_{y^*}^{x^*}] \approx \frac{2\pi}{\lambda(z^*)}\sqrt{\frac{|\det(\nabla^2 H(z^*))|}{\det(\nabla^2 H(x^*))}}\exp\{(H(z^*) - H(x^*))/\epsilon\} \propto \exp\{(H(z^*) - H(x^*))/\epsilon\} \tag{22}$$

where $z^*$ is a unique saddle (that is the maximum of the potential energy barrier) and $\lambda(\cdot)$ denotes the single negative eigenvalue of the Hessian matrix $\nabla^2 H(\cdot)$.

The mean escape time of FLD-based SDE [25] in the continuous space $\mathcal{S} \in \mathbb{R}^N$ is:

$$\mathbb{E}[\tau_{y^*}^{x^*}] \propto w^\alpha \tag{23}$$

where $w$ denotes the "width" of the local minima to the boundary of a potential well.

Similarly, we provide the discrete proposal for the escape time of both LD and FLD. We state upfront that the Markov chains of LD and FLD are reversible if they satisfy the detailed balance conditions. Additionally, $p_\tau(x)$ is a positive stationary distribution, given that the symmetric proposal and the Metropolis-Hastings acceptance criterion are satisfied for constructing discrete LD and FLD. Thus, when the state space is a finite or countable set $\mathcal{S} = \{0, 1\}^N$, the symmetric Dirichlet form:

$$D_\alpha(f, f) = \frac{1}{2}\sum_{x,y\in\mathcal{S}}(f(x) - f(y))^2 p_\tau(x)P_\alpha(x, y) \tag{24}$$

where $P_\alpha(x, y)$ represents the transition matrix. Then, the conductance of an arbitrary non-empty truth subset $\mathcal{B} \subset \mathcal{S}$ is:

$$\Phi_\alpha(\mathcal{B}) = \inf_\mathcal{B}\frac{D_\alpha(\mathbb{1}_\mathcal{B}, \mathbb{1}_\mathcal{B})}{p_\tau(\mathcal{B})} = \frac{1}{p_\tau(\mathcal{B})}\sum_{x\in\mathcal{B}}\sum_{y\notin\mathcal{B}}p_\tau(x)P_\alpha(x, y) \tag{25}$$

The first non-trivial eigenvalue given by the Cheeger inequality of LD [29, 49] is shown as follows:

$$\lambda_1 \geq \frac{(\Phi_2(\mathcal{B}))^2}{2} \tag{26}$$

By the same reasoning, the first non-trivial eigenvalue of FLD [2, 12] is:

$$\lambda_1^\alpha \geq C_{N,\alpha}\Phi_\alpha(\mathcal{B}) = \frac{\alpha 2^{\alpha-1}\Gamma(\frac{N+\alpha}{2})}{\pi^{N/2}\Gamma(1-\frac{\alpha}{2})}\Phi_\alpha(\mathcal{B}) \tag{27}$$

Table 1: Results of compared methods for MIS problem.

| MIS | | RB-[200–300] | | RB-[800–1200] | | ER-[700–800] | | ER-[9000–11000] | |
| --- | --- | --- | --- | --- | --- | --- | --- | --- | --- |
| **Method** | **Type** | **Size ↑** | **Time ↓** | **Size ↑** | **Time ↓** | **Size ↑** | **Time ↓** | **Size ↑** | **Time ↓** |
| Gurobi | OR | 19.98 | 47.57 m | 40.90 | 2.17 h | 41.38 | 50.00 m | — | — |
| KaMIS | OR | 20.10 | 1.40 h | 43.15 | 2.05 h | 44.87 | 52.13 m | 381.31 | 7.60 h |
| DGL | SL | 17.36 | 12.78 m | 34.50 | 23.90 m | 37.26 | 22.71 m | — | — |
| INTEL | SL | 18.47 | 13.07 m | 34.47 | 20.28 m | 34.86 | 6.06 m | 284.63 | 5.02 m |
| DIFUSCO | SL | 18.52 | 16.05 m | — | — | 41.12 | 26.67 m | — | — |
| LTFT | UL | 19.18 | 32 s | 37.48 | 4.37 m | — | — | — | — |
| DiffUCO | UL | 19.24 | 54 s | 38.87 | 4.95 m | — | — | — | — |
| SDDS | UL | 19.62 | 20 s | 39.99 | 6.35 m | — | — | — | — |
| PPO | RL | 19.01 | 1.28 m | 32.32 | 7.55 m | — | — | — | — |
| DIMES | RL | — | — | — | — | 42.06 | 12.01 m | 332.80 | 12.72 m |
| RLNN | PRL | 19.52 | 1.64 m | 38.46 | 6.24 m | 43.34 | 1.37 m | 363.34 | 11.76 m |
| iSCO | H | 19.29 | 2.71 m | 36.96 | 11.26 m | 42.18 | 1.45 m | 365.37 | 1.10 h |
| RLSA | H | 19.97 | 35 s | 40.19 | 1.85 m | 44.10 | 20 s | 375.31 | 1.66 m |
| FLD-IG | PRL | 19.72 | 1.08 m | 39.56 | 6.31 m | 43.50 | 1.35 m | 365.03 | 11.41 m |
| FLD-EG | H | **20.02** | 38 s | **40.25** | 1.93 m | **44.37** | 19 s | **377.50** | 1.12 m |

Derived via spectral expansion, the upper bound of mean escape time for LD and FLD is $\frac{2}{(\Phi_2(\mathcal{B}))^2}$ and $\frac{1}{C_{N,\alpha}\Phi_\alpha(\mathcal{B})}$. Due to the $\Phi_2(\mathcal{B}) \propto \exp\{-\triangle H\}$ [8, 13] and $\Phi_\alpha(\mathcal{B}) \propto w^{-\alpha}$ [26], as the potential barrier $\triangle H$ increases linearly, the escape time of the LD increases exponentially, making it prone to becoming "trapped" at high potential barriers. In contrast, FLD-based SDE, the mean escape time is no longer governed exponentially by the barrier height $\triangle H$ but is instead primarily influenced by the barrier width $w$ in a polynomial manner. Consequently, in multiple-peak landscapes with high potential barriers, FLD-based sampling exhibits superior convergence properties compared to LD.

## 4.4 Enhanced Sampling-Based Approach: FLD-EG

We aim to enhance sampling-based approaches by introducing our **FLD-EG** (i.e., with explicit gradient). As discussed in Sec. 4.3, FLD exhibits a stronger ability to escape from local minima in multiple-peak and high-barrier CO landscapes compared to LD. Motivated by this, we employ FLD-based sampling to guide the assignment of variable values at each iteration. To further enhance stability and reduce the impact of outlier samples that may hinder local exploration, we apply a truncation scheme to remove extreme samples, as illustrated in Fig. 2b. This leads to a more stable and consistent sampling trajectory. Also, inspired by [17], we update only the top-$d$ variables that have the greatest influence on $\nabla H(x)$ and determine the variable values based on the result of the sampling iteration. Specifically, a truncated version of the drift term $\frac{\eta_{n+1}c_\alpha\nabla H(x)}{\tau}$ is applied, guided by a top-$d$ gradient indicator defined as Sigmoid$((\frac{1}{2\tau}((2x-1)\odot\nabla H(x))_i - ((2x-1)\odot\nabla H(x))_{(d)})$. Similarly, the $\mathcal{S}\alpha\mathcal{S}$ noise is truncated based on a top-$d_{\text{noise}}$ noise indicator: Sigmoid$((\frac{1}{2\tau}((2x-1)\odot\nabla H(x))_i - ((2x-1)\odot\nabla H(x))_{(d_{\text{noise}})})$. The final update rule is given in Eq. (19). Details on the algorithmic process can be found in Appendix D.1

Consistent with standard sampling-based approaches, FLD-EG requires a closed-form gradient of the energy function for the CO problems it addresses. Case studies on the energy functions of applied CO problems are presented in Appendix C.

## 4.5 Enhanced Data-Driven Approach: FLD-IG

Since the gradient $\nabla H(x)$ is not available for all CO problems, we propose a data-driven implicit-gradient FLD solver named **FLD-IG**. Inspired by [27] and [17], we introduce an NN-based framework whose training procedure resembles reinforcement learning, alternating between sampling and update steps (we denote this framework as PRL). We introduce the concept of flip probability for variables, as proposed by [17], to mitigate numerical issues, and the network is designed to predict these flip probabilities. Additionally, $\mathcal{S}\alpha\mathcal{S}$ noise is incorporated into the flip decisions at each iteration. To match the FLD process, we first apply a linear transformation to rescale the noise from the range

Table 2: Results of compared methods for MaxCl and MaxCut problems.

| MaxCl | | RB-[200–300] | | RB-[800–1200] | | MaxCut | | BA-[200–300] | | BA-[800–1200] | |
|---|---|---|---|---|---|---|---|---|---|---|---|
| Method | Type | Size ↑ | Time ↓ | Size ↑ | Time ↓ | Method | Type | Size ↑ | Time ↓ | Size ↑ | Time ↓ |
| Gurobi | OR | 19.05 | 1.92 m | 33.89 | 19.67 m | Gurobi | OR | 730.87 | 8.50 m | 2944.38 | 1.28 h |
| ERDOES | UL | 12.02 | 41 s | 25.43 | 2.27 m | ERDOES | UL | 693.45 | 46 s | 2870.34 | 2.82 m |
| LTFT | UL | 16.24 | 42 s | 31.42 | 4.83 m | LTFT | UL | 704.30 | 2.95 m | 2864.61 | 21.33 m |
| DiffUCO | UL | 16.22 | 1.00 m | — | — | DiffUCO | UL | 727.32 | 1.00 m | 2947.53 | 3.78 m |
| SDDS | UL | 18.90 | 38 s | — | — | SDDS | UL | 731.93 | 14 s | **2971.62** | 1.08 m |
| RLNN | PRL | 18.13 | 1.36 m | 35.23 | 7.83 m | RLNN | PRL | 729.00 | 1.58 m | 2907.18 | 3.67 m |
| Greedy | H | 13.53 | 25 s | 26.71 | 25 s | Greedy | H | 688.31 | 13 s | 2786.00 | 3.12 m |
| MFA | H | 14.82 | 27 s | 27.94 | 2.32 m | MFA | H | 704.03 | 1.60 m | 2833.86 | 7.27 m |
| iSCO | H | 18.96 | 54 s | 40.35 | 11.37 m | iSCO | H | 728.24 | 1.67 m | 2919.97 | 4.18 m |
| RLSA | H | **18.97** | 23 s | 40.53 | 1.27 m | RLSA | H | 733.54 | 27 s | 2955.81 | 1.45 m |
| FLD-IG | PRL | 18.52 | 1.40 m | 37.40 | 6.89 m | FLD-IG | PRL | 733.48 | 1.57 m | 2922.54 | 3.07 m |
| FLD-EG | H | **18.97** | 20 s | **40.63** | 1.91 m | FLD-EG | H | **734.18** | 25 s | 2960.13 | 1.70 m |

$[0, 1]$ to $[-1, 1]$ by multiplying $1 - 2x$. The training loss function is defined as:

$$l(\theta; \mathbf{x}, d, \lambda') = \mathbb{E}_{q_\theta(\mathbf{x}_{n+1}|\mathbf{x}_n)}[H(\mathbf{x}_n)] + \lambda' \left[ \sum_{i \in V} q_\theta(\mathbf{x}_{i,n+1}|\mathbf{x}_n) - d \right]^2 \qquad (28)$$

where $q_\theta(\mathbf{x}_{i,n+1}|\mathbf{x}_n)$ still satisfies the mean-field decomposition Eq. (10). Details on the network architecture and the training and inference process can be found in Appendix D.2.

## 5  Experiments

We evaluate our FLD-EG and FLD-IG on three common CO problems, including maximum independent set (**MIS**), maximum clique (**MaxCl**) and max cut (**MaxCut**) problems. Furthermore, we demonstrate more analysis on the sampling trajectories and ablation studies.

**Datasets.** (1) MIS datasets: Following the benchmarks in [17], we evaluate our algorithms on two graph classes: Revised Model B (RB) instances [66] and Erdős–Rényi (ER) random graphs [16] with node weight set to 1; (2) Maximum Clique dataset: we use the single RB graph which is introduced in MIS datasets for the evaluation; (3) Max Cut dataset: we compare our algorithms with the baselines on the Barabási-Albert (BA) graphs [3]. An additional point that warrants special attention is: The size of the training set and the validation set is separately 1000 and 500 graphs for all datasets except for ER-[9000, 11000] (that is, the ER graphs contain 9000 to 11000 nodes), and the test size is 500 for RB and BA graphs; 128 for ER-[700-800] and 16 for ER-[9000, 11000].

**Baselines.** (1) Classical methods: we categorize them into two types: operation research (OR) methods; heuristic (H) methods. In the OR type, we give the general integer linear programming representation of MIS, Maximum Clique, and Max Cut, solved by the Gurobi solver [22] as the OR baseline; especially for MIS, we additionally give the MIS problem-specific solver KAMIS [19]. For the heuristic methods, we give two sampling methods iSCO [51] and RLSA [17] with SA and discrete proposal of LD for all cases; additionally, the Greedy and MFA (Mean-Field Annealing) [6] methods are provided for Maximum Clique and Max Cut problems. (2) Learning-based methods: we classify the learning-based methods into four types: supervised learning (SL), unsupervised learning (UL), reinforcement learning (RL), and pseudo reinforcement learning (PRL) which uses the sample and update scheme similar to the RL. For SL, the INTEL with GCN and probability heatmap [33], DGL with Monte Carlo Tree Search and two GNN backbones [7], and DIFUSCO with UNet-Style diffusion model [53] are given as SL baselines for the MIS problem. In the type of UL, the LIFT with GFlownets [68], DiffUCO with UNet-Style diffusion model [44], and SDDS with discrete diffusion models [43] are presented for three case studies; beyond that, ERDOES with random graph model [16] is set for the case studies without the MIS problem. The RL methods, PPO with actor-and-critic [1] and DIMES with reinforcement optimization combined with meta-learning [40], are introduced for the MIS problem. Similar to the FLD-IG, RLNN with the discrete proposal of LD is the baseline method of the PRL for three case studies [17].

**Main Results.** In the main experiments for the evaluation of our FLD-EG and FLD-IG, we give two metrics: the objective value of each problem and the overall sequential testing time, which

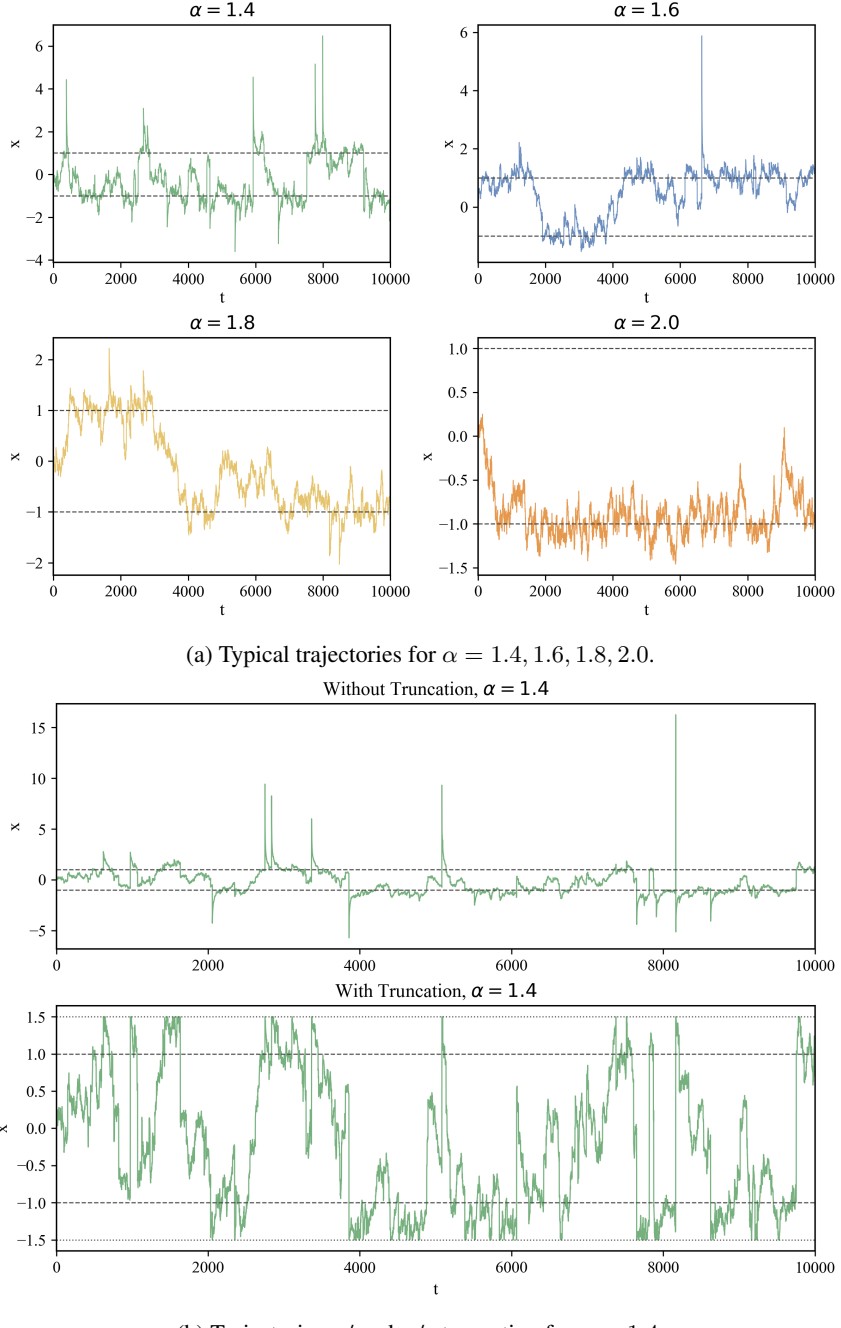

(a) Typical trajectories for $\alpha = 1.4, 1.6, 1.8, 2.0$.

(b) Trajectories w/ and w/o truncation for $\alpha = 1.4$.

Figure 2: Sampling trajectories of the FLD-based SDE.

attracted the main concern in the field of CO. For three case studies in this work, the objective values separately represent the independent set size of MIS, the clique size of Maximum Clique, and the cut size of the Max Cut (the detailed formation can be seen in Appendix C). The reported results for each learning-based method correspond to the longest runtimes and, accordingly, should also exhibit the highest objective values; with regard to heuristic methods, we fixed the number of iterations to be the same. Table 1 and 2 demonstrate the results on MIS, MaxCl and MaxCut. On most problems, our FLD-EG and FLD-IG outperform the SOTA classical and learning-based methods, respectively, achieving higher objective values in shorter or comparable runtimes. For the classical methods, since OR methods can obtain the optimal solution given sufficient runtimes, we report the results of

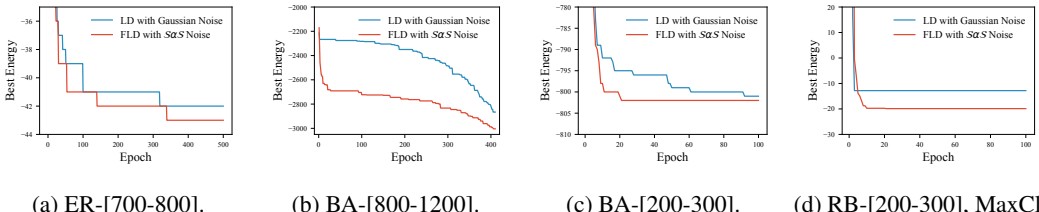

(a) ER-[700-800].  (b) BA-[800-1200].  (c) BA-[200-300].  (d) RB-[200-300], MaxCl.

Figure 3: Ablation for our methods (FLD with $\mathcal{S}\alpha\mathcal{S}$) and LD sampling process method with Gaussian noise. The staircase curves show how the "Best Energy" evolves as the number of iterations ("epochs"), where the "Best Energy" is the minimum energy function value between last and current epoch.

OR methods solely to demonstrate the performance gap between the heuristic and learning-based methods relative to the optimum, rather than to make a direct comparison. In the competitive heuristic methods (i.e. iSCO, RLSA and our FLD-EG), our FLD-EG can attain equal or superior objective values within the same sampling steps, while maintaining comparable or slightly reduced runtime. In the field of learning-based methods, our FLD-IG achieves higher objective values than the others on 75% of datasets; among methods with comparable performance metrics, our FLD-IG achieves shorter runtimes on all datasets except the RB-[800-1200] instance.

**Sampling Trajectories.** We conduct FLD-based iterative sampling under different values of $\alpha$ to simulate the trajectory of a single variable $x$ during CO solving. Owing to the symmetry of the $\mathcal{S}\alpha\mathcal{S}$, we shift and scale $x$ from [0, 1] to [-1, 1] in our methods. As shown in Fig. 2a, when $\alpha = 2$ (i.e., the process degenerates to LD sampling), $x$ readily becomes trapped in local minima, leading to slow convergence; as $\alpha$ decreases—intensifying the heavy tail of the $\mathcal{S}\alpha\mathcal{S}$ distribution—$x$ flips more frequently between –1 and 1, promoting escape from local minima and accelerating convergence (theoretical justification is provided in Sec. 4.3). However, for very small $\alpha$, the generation of large outliers can cause sampled points to deviate excessively, losing track of the underlying landscape. To remedy this, we introduce a truncation strategy. As shown in Fig. 2b, bounding the sampled points within a prescribed range yields a markedly more stable sampling process.

**Ablation Studies.** To rigorously validate the effectiveness of our $\mathcal{S}\alpha\mathcal{S}$-noise sampling process, we conducted ablation studies comparing its convergence behavior against that of LD with Gaussian noise when solving CO problems (cf. Sec. 4.3). Specifically, in both FLD-EG and FLD-IG, we replaced the FLD sampling process driven by $\mathcal{S}\alpha\mathcal{S}$ noise with the LD sampling process driven by Gaussian noise, and designed comparative experiments to monitor the iterative descent of the energy function. As shown in Fig. 3, two panels on the left depict the ablation studies for FLD-EG, while the others depict the ablation studies for FLD-IG. Notably, since the optimal solution is not attained, there is a gap between the best energy function value and the current objective value, which corresponds to the penalty term imposed by the constraints (i.e. $\lambda b(x)$ shown in Eq. (6)). The results in the figure indicate that, whether using explicit gradient or implicit gradient, our method markedly outperforms LD with Gaussian noise; it not only demonstrates superior ability of FLD with $\mathcal{S}\alpha\mathcal{S}$ noise to escape local optima compared to LD, but also its capacity to converge to a lower energy function value.

## 6 Conclusion and Outlook

In this paper, we have addressed the limitations of LD in CO, including its tendency to be trapped in local optima, slow convergence, and generally suboptimal iterative performance. To overcome these challenges, we propose a FLD sampling process driven by $\mathcal{S}\alpha\mathcal{S}$ noise, fortified with a truncation strategy to ensure sampling stability. We theoretically prove that FLD achieves a polynomial mean escape time—significantly faster than the exponential escape time of LD, which depends on the energy barrier height—thereby enabling more rapid convergence. By integrating FLD into both sampling-based and data-driven frameworks, accommodating problems with or without explicit gradient information, we demonstrate its superior convergence and exploration capabilities on three case studies: MIS, MaxCl and MaxCut. Our FLD-EG and FLD-IG achieve SOTA or near-SOTA results compared to existing methods. Although our current FLD design focuses on binary-variable CO problems, it has potential applicability to integer and continuous-variable formulations. We plan to investigate these promising extensions in future work.

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

## A    Proof of Theorem 1

**Theorem 4** ([47]). *Consider the SDE* (13), *the drift b is defined by:*

$$b(x,\alpha) = (\mathcal{D}^{\alpha-2} f_\pi(x))/\phi(x) \tag{29}$$

$f_\pi(x) = -\phi(x)\partial_x U(x)$, *and $\pi$ is an invariant measure of the Markov process.*

**Theorem 5** ([38]). *The Riesz derivative $\mathcal{D}^\gamma$ of a function $f(x)$ can be defined as the limit of the fractional centered difference operator $\triangle_h^\gamma$, given:*

$$\mathcal{D}^\gamma f(x) = \lim_{h\to 0} \triangle_h^\gamma f(x) \tag{30}$$

*where,*

$$\triangle_h^\gamma f(x) = (1/h^\gamma) \sum_{k=-\infty}^{\infty} g_{\gamma,k} f(x - kh) \tag{31}$$

*and*

$$g_{\gamma,k} = (-1)^k \frac{\Gamma(\gamma+1)}{\Gamma(\gamma/2 - k + 1)\Gamma(\gamma/2 + k + 1)} \tag{32}$$

*Proof.* Theorem 4 guarantees the existence of the equality on the left-hand side of Eq.(13).

Next, we will give the proof of the right-hand side. The more computationally efficient variant of the first numerical scheme presented in Theorem 5 is given as follows [9]:

$$\mathcal{D}^\gamma f_\pi(x) \approx g_{\gamma,0} f_\pi(x) \tag{33}$$

where $g_{\gamma,0} = \Gamma(\gamma+1)/\Gamma(\gamma/2+1)^2$ for $x \in \mathbb{R}$.

Then we get

$$b(x,\alpha) = \frac{\mathcal{D}^{\alpha-2} f_{p_\tau}(x)}{\phi(x)} = \frac{\mathcal{D}^{\alpha-2}(-\phi(x)\partial_x H(x))}{\phi(x)} = \mathcal{D}^{\alpha-2} s(x)$$
$$= \Gamma(\alpha-1)/\Gamma(\alpha/2)^2 s(x) = c_\alpha s(x) \tag{34}$$

where $c_\alpha = \Gamma(\alpha-1)/\Gamma(\alpha/2)^2$. □

## B    Proof of Theorem 2 and Theorem 3

**Definition 4** ([64]). *A random variable $X$ is $\alpha$-stable if and only if its characteristic function is given by*

$$\log\phi(t) = \begin{cases} -\sigma_2^\alpha |t|^\alpha \exp\{-i\beta_2 \text{sign}(t)\frac{\pi}{2} K(\alpha)\} + i\mu t, \ \alpha \neq 1 \\ -\sigma_2 |t|(\frac{\pi}{2} + i\beta_2 \text{sign}(t)\log|t|) + i\mu t, \ \alpha = 1 \end{cases} \tag{35}$$

*where, $K(\alpha) = \alpha - 1 + \text{sign}(1-\alpha)$, and $\text{sign}(\cdot)$ denotes the sign function. The parameters $\sigma_2$ and $\beta_2$ are related to $\sigma$ and $\beta$.*

***Case 1:*** *for $\alpha \neq 1$, $\beta_2$ is such that*

$$\tan\left(\beta_2 \frac{\pi K(\alpha)}{2}\right) = \beta \tan\frac{\pi\alpha}{2} \tag{36}$$

*and the new scale parameter*

$$\sigma_2 = \sigma\left(1 + \beta^2 \tan^2\frac{\pi\alpha}{2}\right)^{\frac{1}{2\alpha}} \tag{37}$$

***Case 2:*** *for $\alpha = 1$, $\beta_2 = \beta$ and $\sigma_2 = \frac{2}{\pi}\sigma$.*

**Lemma 1** ([64]). *$X$ is a $\mathcal{S}_\alpha(1,\beta_2,0)$ random variable if and only if for $x > 0$:*

$$\begin{cases} P(0 < X \leq x) = \dfrac{1}{\pi}\displaystyle\int_{\gamma_0}^{\frac{\pi}{2}} \exp\left[-x^{\frac{\alpha}{\alpha-1}} U_\alpha(\gamma,\gamma_0)\right] d\gamma, \ \alpha < 1 \\ P(X \geq x) = \dfrac{1}{\pi}\displaystyle\int_{\gamma_0}^{\frac{\pi}{2}} \exp\left[-x^{\frac{\alpha}{\alpha-1}} U_\alpha(\gamma,\gamma_0)\right] d\gamma, \ \alpha > 1 \end{cases} \tag{38}$$

*where, $\gamma_0 = -\frac{\pi\beta K(\alpha)}{2\alpha}$ and $U_\alpha(\gamma,\gamma_0) = \left(\frac{\sin\alpha(\gamma-\gamma_0)}{(\cos\gamma)}\right)^{\alpha/1-\alpha} \frac{\cos(\gamma-\alpha(\gamma-\gamma_0))}{\cos\gamma}$.*

*Proof.* When $\gamma > \gamma_0$ then the right hand side of Eq. (17) ($\alpha \neq 1$) is positive and can be expressed as [64]:

$$\left(\frac{a(\gamma)}{W}\right)^{(1-\alpha)/\alpha},\tag{39}$$

where

$$a(\gamma) = \left(\frac{\sin\left(\alpha(\gamma - \gamma_0)\right)}{\cos\gamma}\right)^{\alpha/(1-\alpha)}\frac{\cos\left(\gamma - \alpha(\gamma - \gamma_0)\right)}{\cos\gamma}.\tag{40}$$

**Case 1:** When $0 < \alpha < 1$, Eq. (17) ($\alpha \neq 1$) implies that $X > 0$ if and only if $\gamma > \gamma_0$. Since $\frac{1-\alpha}{\alpha} > 0$, we can write

$$
\begin{aligned}
P(0 < X \leq x) &= P\left(0 < X \leq x,\ \gamma > \gamma_0\right)\\
&= P\left(0 < \left(a(\gamma)/W\right)^{(1-\alpha)/\alpha} \leq x,\ \gamma > \gamma_0\right)\\
&= P\left(W \geq x^{\alpha/(\alpha-1)}\,a(\gamma),\ \gamma > \gamma_0\right)\\
&= E_\gamma\left[\exp\left(-x^{\alpha/(\alpha-1)}\,a(\gamma)\right)\mathbf{1}_{\{\gamma>\gamma_0\}}\right]\\
&= \frac{1}{\pi}\int_{\gamma_0}^{\pi/2}\exp\left(-x^{\alpha/(\alpha-1)}\,a(\gamma)\right)d\gamma.
\end{aligned}
$$

From Lemma 1, we conclude that $X \sim S_\alpha(1, \beta_2, 0)$.

**Case 2:** For $1 < \alpha \leq 2$, noting that $\frac{\alpha-1}{\alpha} > 0$, we similarly deduce that for all $x > 0$,

$$
\begin{aligned}
P(X \geq x) &= P\left(X \geq x,\ \gamma > \gamma_0\right)\\
&= P\left(\left(a(\gamma)/W\right)^{(1-\alpha)/\alpha} \geq x,\ \gamma > \gamma_0\right)\\
&= P\left(\left(W/a(\gamma)\right)^{(\alpha-1)/\alpha} \geq x,\ \gamma > \gamma_0\right)\\
&= P\left(W \geq x^{\alpha/(\alpha-1)}\,a(\gamma),\ \gamma > \gamma_0\right)\\
&= E_\gamma\left[\exp\left(-x^{\alpha/(\alpha-1)}\,a(\gamma)\right)\mathbf{1}_{\{\gamma>\gamma_0\}}\right]\\
&= \frac{1}{\pi}\int_{\gamma_0}^{\pi/2}\exp\left(-x^{\alpha/(\alpha-1)}\,a(\gamma)\right)d\gamma.
\end{aligned}
$$

Again by Lemma 1 we get $X \sim S_\alpha(1, \beta_2, 0)$.

**Case 3:** For the case $\alpha = 1$, when $\beta_2 = 0$ the right hand side of Eq. (17) ($\alpha = 1$) simplifies to $\frac{\pi}{2}\tan\gamma$, which has a Cauchy law (cf. Eq. (35)). If $\beta_2 \neq 0$, it can instead be written as

$$\beta_2 \log\left(\frac{a_1(\gamma)}{W}\right),\tag{41}$$

where

$$a_1(\gamma) = \frac{\frac{\pi}{2} + \beta_2\gamma}{\cos\gamma}\ \exp\left(\frac{1}{\beta_2}\left(\tfrac{\pi}{2} + \beta_2\gamma\right)\tan\gamma\right).\tag{42}$$

For $\beta_2 > 0$, we have

$$
\begin{aligned}
P(X \leq x) &= P\left(\beta_2 \log(a_1(\gamma)/W) \leq x\right)\\
&= P\left(W \geq e^{-x/\beta_2}\,a_1(\gamma)\right)\\
&= E_\gamma\left[\exp\left(-e^{-x/\beta_2}\,a_1(\gamma)\right)\right]\\
&= \frac{1}{\pi}\int_{-\pi/2}^{\pi/2}\exp\left(-e^{-x/\beta_2}\,a_1(\gamma)\right)d\gamma.
\end{aligned}
$$

Finally, we conclude that for all $\beta_2$, $X \sim S_1(1, \beta_2, 0)$.

Due to the $\mathcal{S\alpha S}$ distribution being the special case of $\alpha$-stable distribution when $\beta = 0$, thus Theorem 2 and Theorem 3 have been proven together. $\qquad\square$

# C  Case Study on Energy Functions

The problem formulations utilized in this paper are given below for three common case studies with the closed form of the energy function $H(x)$, which means $H(x)$ is first-order derivable. Let $G(V, E)$ be an undirected and unweighted graph, where $V = \{1, 2, \cdots, N\}$ denotes the node set of the graph $G$ and $E \subseteq V \times V$ represents the edge set. The problem descriptions, energy function, and its gradient are given.

**Case Study 1: Maximum Independent Set (MIS).** The independent set is $S$ satisfied that $\forall i, j \in S \subseteq V$ and $e(i, j) \in E$, then $i = j$. Thus the definition of MIS $S^* = \arg\max_{S \subseteq V} |S|$ ($|\cdot|$ denotes the size of $\cdot$). The formulation of MIS is:

$$\max_{x \in \{0,1\}^N} \sum_{i \in V} c_i x_i, \text{ s.t. } \sum_{e(i,j) \in E} x_i x_j = 0 \tag{43}$$

The energy function of MIS formed as Eq. (6) is:

$$H(x) = -\sum_{i \in V} c_i x_i + \lambda \sum_{e(i,j) \in E} x_i x_j = -\mathbf{c}^\top \mathbf{x} + \lambda \frac{\mathbf{x}^\top \mathbf{A} \mathbf{x}}{2} \tag{44}$$

where $c_i$ $(i \in V)$ denotes the node weights of graph $G$ and the content to the right of the last equal sign is the energy function under the matrix form representation (the same goes for the following two cases). The gradient of the energy function can be presented readily:

$$\nabla H(x) = -\mathbf{c} + \lambda \mathbf{A} \mathbf{x} \tag{45}$$

**Case Study 2: Maximum Clique.** The clique is the set $C \subseteq V$ satisfied that $\forall i, j \in C$, $i \neq j$, then $e(i, j) \in E$. Therefore, the definition of maximum clique $C^* = \arg\max_{S \subseteq V} |C|$. The formulation of maximum clique is:

$$\max_{x \in \{0,1\}^N} \sum_{i \in V} c_i x_i, \text{ s.t. } \sum_{e(i,j) \notin E} x_i x_j = 0 \tag{46}$$

The energy function of maximum clique formed as Eq. (6) is:

$$H(x) = -\sum_{i \in V} c_i x_i + \lambda \sum_{e(i,j) \notin E} x_i x_j = -\mathbf{c}^\top \mathbf{x} + \frac{\lambda}{2}((\mathbf{1}^\top x)^2 - \mathbf{x}^\top \mathbf{x} - \mathbf{x}^\top \mathbf{A} \mathbf{x}) \tag{47}$$

The gradient of $H(x)$ is shown as:

$$\nabla H(x) = -\mathbf{c} + \lambda((\mathbf{1}^\top \mathbf{x})\mathbf{1} - \mathbf{x} - \mathbf{A} \mathbf{x}) \tag{48}$$

**Case Study 3: Max Cut.** The max cut problem seeks a partition $(S, \bar{S})$ that maximizes the number of crossing edges:

$$\max_{S \subseteq V} |\{e(i, j) \in E | i \in S, j \in \bar{S} = V \setminus S\}| \tag{49}$$

The mathematical formulation, energy function, and its gradient can be presented as:

$$\max_{x \in \{0,1\}^N} \sum_{e(i,j) \in E} \frac{1 - (2x_i - 1)(2x_j - 1)}{2} \tag{50}$$

$$H(x) = -\sum_{e(i,j) \in E} \frac{1 - (2x_i - 1)(2x_j - 1)}{2} = \mathbf{x}^\top \mathbf{A} \mathbf{x} - \mathbf{1}^\top \mathbf{A} \mathbf{x} \tag{51}$$

$$\nabla H(x) = \mathbf{A}(2\mathbf{x} - \mathbf{1}) \tag{52}$$

# D   Details on FLD-EG and FLD-IG

## D.1   Algorithmic Process of FLD-EG

We now present the detailed implementation of the FLD-EG algorithm in Alg. 1. The algorithm takes as input the maximum number of iterations $T$, the number $K$ of independent SA processes, the truncation parameter $d$ for the closed-form gradient $\nabla H$, the truncation parameter $d_{\mathrm{noise}}$ for the $\mathcal{S}\alpha\mathcal{S}$ noise, the initial temperature $\tau_0$ for SA, the stability parameter $\alpha$ of the FLD sampling process, and the stepsize schedule parameters $a_\eta$ and $b_\eta$. (For the specific values of these hyperparameters, see Appendix E.2.)

The FLD-EG algorithm employs a near–continuous sampling procedure to guide the assignment of binary variables. First, both the binary variable vector $\mathbf{x}$ and the auxiliary continuous variable vector $\mathbf{h}$ are initialized, and the initial energy is computed via the problem-specific function energy_func(). Each iteration then comprises two stages: (1) a sampling update for $\mathbf{h}$, and (2) an update for $\mathbf{x}$ based on the newly sampled $\mathbf{h}$. After each update of $\mathbf{x}$, we record the best observed energy. Once the maximum iteration count $T$ is reached, a greedy decoding step produces a final, constraint-satisfying solution.

The sampling update for $\mathbf{h}$ consists of four substeps:

1. **Noise sampling.** Sample the noise variable $z\_iter$ from an $\mathcal{S}\alpha\mathcal{S}(1)$ distribution as prescribed by Theorem 3, with the sampling mechanism defined in Eq. (18). In practice, for convenience, we replace exponentially distributed sampling with uniformly distributed sampling for $U$.

2. **Gradient truncation.** Apply the Top-$d$ truncated indicator to $\nabla H$, ensuring that only the $d$ components with the largest magnitude influence the update.

3. **Noise truncation.** Apply the Top-$d_{\mathrm{noise}}$ truncated indicator to the sampled noise vector, truncating extreme values to stabilize the sampling process.

4. **State update.** Update $\mathbf{h}$ according to the FLD update rule in Eq. (19).

Finally, since the solution may still violate some problem constraints, we perform a greedy decoding step on $\mathbf{x}$ until all constraints are satisfied.

## D.2   Details on FLD-IG

**Training and Inference Process.** In Alg. 2, we present the detailed training procedure for FLD-IG. The algorithm takes as input the maximum number of iterations $T'$ for training and $T$ for inference, the number of independent parallel sampling processes $K'$ for training and $K$ for inference, the truncation parameter $d$ for both the gradient and the $\mathcal{S}\alpha\mathcal{S}$ noise, the initial temperature $\tau_0$ for the sampling process, and the stability parameter $\alpha$ of the FLD sampling process. (For the specific values of these hyperparameters, see Appendix E.2.) At each iteration, we sample $\mathcal{S}\alpha\mathcal{S}$ noise and incorporate it into the proposal distribution, allowing $\mathbf{x}'$ to be drawn from this noise-augmented proposal so as to compute flip probabilities and update $\mathbf{x}$. Unlike FLD-EG, which applies truncation to both the gradient and the noise terms during sampling, and given that the noise samples are independent at each iteration and no closed-form formulation exists for the energy-function gradient, we instead perform a unified truncation of the noise-augmented proposal distribution directly within the loss function. After each update, we record the best-observed energy. Once the maximum training iteration step $T'$ is reached, save the best parameters for the inference process.

Similar to the training process, we first sample the $\mathcal{S}\alpha\mathcal{S}$ noise and incorporate it into the proposal distribution in the inference process. Next, we draw $\mathbf{x}'$ by sampling from this noise-augmented proposal distribution to compute flip probabilities for updating $\mathbf{x}'$, and we record the best energy observed after each update. Once the maximum inference iteration count $T$ is reached, a final greedy decoding step generates a constraint-satisfying solution which is similar to the greedy decoding step of FLD-EG.

**Network Architecture.** For the implementation of the network architecture, we adopt a five-layer GCN with 128 hidden dimensions, which is consistent with [17]. We observe that our FLD-IG converges faster than [17]. Therefore, we set the number of training epochs to 30, which is adequate for convergence.

**Algorithm 1** FLD-EG
***
**Require:** $T, K, d, d_{\text{noise}}, \tau_0, \alpha, a_\eta,$ and $b_\eta$.

1: Initialize $\mathbf{x} \in \{0, 1\}^{N \times k}$;    $h \leftarrow \mathbf{x}$;    $\mathbf{x}^* \leftarrow \mathbf{x}$;    $c_\alpha = \Gamma(\alpha - 1)/\Gamma(\alpha/2)^2$

2: Build adjacency matrix $A$ from (edge_index, edge_weight)

3: $(\text{energy}, \nabla H) \leftarrow \text{energy\_func}(A, b, \mathbf{x}, True)$

4: **best\_sol** $\leftarrow \mathbf{x}$;    **best\_energy** $\leftarrow$ energy

5: **for** $t = 1, 2, \cdots, T$ **do**

6:    $\tau \leftarrow \tau_0 \left(1 - \dfrac{t}{T}\right)$

7:    **for** $i = 1, 2, \cdots, N$ **do**

8:      Sample $W \sim \mathcal{U}(-\frac{\pi}{2}, \frac{\pi}{2})$,    $U \sim \mathcal{U}(0, 1)$

9:      $z\_iter \leftarrow \dfrac{\sin(\alpha W)}{\cos(W)^{1/\alpha}} \left(\dfrac{\cos((1 - \alpha)W)}{-\ln U}\right)^{\frac{1-\alpha}{\alpha}}$           $\triangleright \mathcal{S}\alpha\mathcal{S}$ Noise Sampling

10:      $\eta \leftarrow \left(\dfrac{a_\eta}{t + 1}\right)^{b_\eta}$

11:      $t_g \leftarrow -\text{kthvalue}(-\frac{1}{2}\left((2\mathbf{x} - 1) \odot \nabla H\right)_{(i)}, d, \dim = 0)$    $\triangleright$ Top-$d$ Truncated Indicator

12:      $p_g \leftarrow \text{Sigmoid}\left((\frac{1}{2}\left((2\mathbf{x} - 1) \odot \nabla H\right)_{(i)} - t_g)/\tau\right)$

13:      Sample $\mathbf{I}_g \sim \text{Bernoulli}(p_g)$

14:      $t_z \leftarrow -\text{kthvalue}(-\frac{1}{2}\left((2\mathbf{x} - 1) \odot \nabla H\right)_{(i)}, d_{\text{noise}}, \dim = 0)$    $\triangleright$ Top-$d_{\text{noise}}$ Truncated Indicator

15:      $p_z \leftarrow \text{Sigmoid}\left((-\frac{1}{2}\left((2\mathbf{x} - 1) \odot \nabla H\right)_{(i)} - t_z)/\tau\right)$

16:      Sample $\mathbf{I}_z \sim \text{Bernoulli}(p_z)$

17:      $grad\_iter \leftarrow \eta \, c_\alpha \left(-\frac{1}{\tau}\right) \nabla H$

18:      $z\_iter \leftarrow \eta^{1/\alpha} z\_iter$

19:      $h_i \leftarrow h_i \, - \, \mathbf{I}_g \odot grad\_iter \, + \, \mathbf{I}_z \odot z\_iter$        $\triangleright$ Sampling Iterative Process

20:      $h_i \leftarrow \text{Clamp}(h_i, 0, 1)$

21:      $x_i \leftarrow \text{where}\left(\text{rand}() < h_i, 0, 1\right)$

22:    **end for**

23:    $(\text{energy}, \nabla H) \leftarrow \text{energy\_func}(A, b, \mathbf{x}, epoch < T)$

24:    **if** energy $<$ **best\_energy then**

25:      **best\_sol** $\leftarrow \mathbf{x}$;    **best\_energy** $\leftarrow$ energy

26:    **end if**

27: **end for**

28: **return best\_sol**    (**or** return min **best\_energy** if skip-decode)
***

# E   Details on Experiments

## E.1   Hardware and Software Devices

Experiments are conducted on a Linux workstation using an H100 GPU and an Intel(R) Xeon(R) Platinum 8468 CPU, with programs implemented in *PyTorch*.

## E.2   Hyperparameters

We show the utilized hyperparameter values of FLD-EG and FLD-IG in Table 3 and Table 4, respectively. The selection of hyperparameter values partly refers to [17].

# F   Broader Impacts

The FLD framework we introduce has the potential to substantially advance both the practical application and theoretical understanding of CO. By enabling reliable escape from deep local optima—and doing so in polynomial time across a range of problem landscapes—FLD can drive more efficient logistic networks, reducing transportation costs and carbon emissions through better routing; streamline scheduling in manufacturing and cloud computing, leading to higher resource utilization

**Algorithm 2** FLD-IG (Training)

---

**Require:** $T'$, $K'$, $d$, $\lambda'$, $\alpha$, $\tau_0$
1: Initialize $\theta$
2: **while** stopping criterion not met **do**
3:    Initialize $x \in \{0,1\}^N$,    $\mathcal{D} \leftarrow \{x\}$
4:    **for** $t = 1, 2, \cdots, T'$ **do**
5:       $\tau \leftarrow \tau_0 \left(1 - \frac{t-1}{T'}\right)$
6:       Sample $\mathbf{W}_{(i)} \overset{\text{i.i.d.}}{\sim} \mathcal{U}\!\left(-\frac{\pi}{2}, \frac{\pi}{2}\right)$,    $i = 1, \ldots, N$
7:       Sample $\mathbf{U}_{(i)} \overset{\text{i.i.d.}}{\sim} \mathcal{U}(0,1)$,    $i = 1, \ldots, N$
8:       Compute

$$\mathbf{Z}_{(i)} \leftarrow \frac{\sin(\alpha \mathbf{W}_{(i)})}{\cos(\mathbf{W}_{(i)})^{1/\alpha}} \left( \frac{\cos((1-\alpha)\mathbf{W}_{(i)})}{-\ln \mathbf{U}_{(i)}} \right)^{\frac{1-\alpha}{\alpha}}, \quad i = 1, \ldots, N$$

$\triangleright \mathcal{S}\alpha\mathcal{S}$ Noise Sampling
9:       $\mathbf{x}' \sim q_\theta\big(\mathbf{x}' \mid \mathbf{x} + \tau \mathbf{Z}(1 - 2\mathbf{x})\big)$       $\triangleright$ Sample from proposal distribution with $\mathcal{S}\alpha\mathcal{S}$ Noise
10:       $\mathcal{D} \leftarrow \mathcal{D} \cup \{x'\}$
11:       $x \leftarrow x'$
12:    **end for**
13:    $\theta \leftarrow \arg\min_\theta \mathbb{E}_{x \in \mathcal{D}}\big[\ell_{\text{FLD}}(\theta; x, d, \lambda', \alpha, \tau_0)\big]$
14: **end while**
15:
16: **return** $\theta$

---

Table 3: Hyperparameters used by FLD-EG on all datasets.

| Problem | Dataset | $\tau_0$ | $d$ | $K$ | $T$ | $\lambda$ | $d_{\text{noise}}$ | $\alpha$ | $a_\eta$ | $b_\eta$ |
|---|---|---|---|---|---|---|---|---|---|---|
| MIS | RB-[200–300] | 0.01 | 5 | 200 | 300 | 1.02 | 9 | 1.2 | 4 | 0.6 |
| | RB-[800–1200] | 0.01 | 5 | 200 | 500 | 1.02 | 12 | 1.3 | 0.1 | 0.6 |
| | ER-[700–800] | 0.01 | 20 | 200 | 500 | 1.001 | 20 | 1.5 | 0.1 | 0.6 |
| | ER-[9000–1100] | 0.01 | 20 | 200 | 5000 | 1.001 | 60 | 1.1 | 0.1 | 0.6 |
| MaxCl | RB-[200–300] | 4 | 2 | 200 | 100 | 1.02 | 4 | 1.5 | 30 | 0.6 |
| | RB-[800–1200] | 4 | 2 | 200 | 500 | 1.02 | 2 | 1.7 | 0.1 | 0.6 |
| MaxCut | BA-[200–300] | 5 | 20 | 200 | 200 | 1.02 | 33 | 1.01 | 200 | 0.6 |
| | BA-[800–1200] | 5 | 20 | 200 | 500 | 1.02 | 35 | 1.01 | 200 | 0.6 |

Table 4: Hyperparameters used by FLD-IG on all datasets.

| Problem | Dataset | $\tau_0$ | $d$ | $K$ | $T$ | $\lambda$ | $K'$ | $T'$ | $\lambda'$ | $\alpha$ |
|---|---|---|---|---|---|---|---|---|---|---|
| MIS | RB-[200–300] | 0.25 | 5 | 20 | 100 | 1.02 | 10 | 50 | 0.5 | 1.7 |
| | RB-[800–1200] | 0.25 | 5 | 20 | 200 | 1.02 | 10 | 300 | 0.5 | 1.7 |
| | ER-[700–800] | 0.6 | 20 | 20 | 200 | 1.001 | 10 | 500 | 0.5 | 1.7 |
| | ER-[9000–1100] | 0.9 | 20 | 20 | 800 | 1.001 | – | – | – | 1.7 |
| MaxCl | RB-[200–300] | 0.25 | 2 | 20 | 100 | 1.02 | 10 | 100 | 0.5 | 1.7 |
| | RB-[800–1200] | 0.25 | 2 | 20 | 200 | 1.02 | 10 | 300 | 0.5 | 1.7 |
| MaxCut | BA-[200–300] | 0.25 | 20 | 20 | 100 | 1.02 | 10 | 50 | 0.5 | 1.7 |
| | BA-[800–1200] | 0.25 | 20 | 20 | 200 | 1.02 | 10 | 300 | 0.5 | 1.7 |

and energy savings; and enhance network-design and financial-optimization tools, yielding more robust communication infrastructures and investment strategies. Moreover, because FLD naturally integrates with data-driven pipelines via its implicit-gradient formulation, it can be seamlessly

incorporated into emerging machine-learning platforms for applications such as automated materials discovery, bioinformatics, and large-scale resource allocation, fostering interdisciplinary innovation.

At the same time, we acknowledge that any powerful optimization technology carries risks. Unchecked, FLD could be used to accelerate adversarial planning—such as in cybersecurity, market manipulation, or autonomous weaponry—by quickly finding worst-case configurations. To mitigate misuse, we recommend that practitioners pair FLD with domain-specific ethical guidelines and transparency mechanisms (e.g., logging and audit trails for critical decision systems), and that the research community pursue formal verification methods to ensure that FLD-based solutions adhere to safety and fairness constraints. By proactively addressing these considerations, we believe FLD can serve as a force for positive impact—improving efficiency and sustainability in industrial and scientific applications while minimizing potential harms.

