# OpenReview forum: "Fractional Langevin Dynamics for Combinatorial Optimization via Polynomial-Time Escape"
_NeurIPS.cc/2025/Conference — NeurIPS 2025 poster_

### Official Review · Reviewer_kHZa · 2025-06-01

**Clarity:** 3
**Significance:** 3
**Originality:** 3
**Rating:** 4
**Confidence:** 2

**Summary:**

This paper studies combinatorial optimization and Langevin Dynamics and proposes FLD for CO.

Theoretically the authors demonstrate that their algorithm enhances exploration and improves convergence.

Empirically, results on several problems (the Maximum Independent Set, Maximum Clique, and Maximum Cut problems) further demonstrate that incorporating FLD advances existing approaches and they achieve SOTA performance in most of the experiments.

**Questions:**

Could you make parentheses the big ones in equation (2), line 76?

The parentheses are not in the correct form in other places, e.g., line 111.

The topic of this paper is not very well align with my expertise, therefore I'm not able to evaluate it with much confidence.

**Ethical Concerns:**

["NO or VERY MINOR ethics concerns only"]

**Final Justification:**

The authors have addressed the questions I raised in my review.

After checking the other reviews and the rebuttal, I believe this work makes a solid contribution.

**Paper Formatting Concerns:**

No formatting issues

**Quality:**

3

**Strengths And Weaknesses:**

Strengths:

This paper is well structured, clearly written, and easy to follow.

This paper presents both solid algorithmic design, theoretical proof and insights, plus comprehensive empirical results.

Weaknesses:

The related work part seems to be too concise. It could be extended a little bit.

---

> ### Author Rebuttal · Authors · 2025-07-30
>
> We sincerely appreciate the effort you put into reviewing our paper. Below, we provide our responses to the weakness and questions you raised.
>
> > **W1: The related work part seems to be too concise. It could be extended a little bit.**
>
> Thank you for your suggestion. We originally intended to include the Related Work in the main text, but due to space constraints we only provided a partial discussion and subsequently moved it **to Appendix A**. Unfortunately, submission time pressure prevented us from expanding it further. We apologize for this oversight and will fully supplement the Related Work section in the final version of our paper to provide a more comprehensive background review.
>
> **To Q1&Q2**: Thank you for catching this error. We will make the corrections in the final version of our paper and include the correct version in the final submission.
>
> If you have any additional questions, please don't hesitate to reach out. We would be more than happy to address any concerns.

---

> > ### Comment · Reviewer_kHZa · 2025-08-04
> >
> > Thanks for addressing my questions! I believe this work makes a significant contribution!

---

> > > ### Author Response · Authors · 2025-08-05
> > >
> > > Thank you for reviewing our rebuttal. We’re glad the clarifications addressed your concerns. Please let us know if anything further would be helpful.

---

### Official Review · Reviewer_ymVV · 2025-06-28

**Clarity:** 3
**Significance:** 3
**Originality:** 3
**Rating:** 4
**Confidence:** 4

**Summary:**

This paper introduces ​​Fractional Langevin Dynamics (FLD)​​ to address key limitations of traditional Langevin Dynamics (LD) in combinatorial optimization (CO). LD's reliance on Gaussian noise hinders its ability to escape deep local optima, requires expensive parallel chains, and struggles with rugged landscapes or strict constraints. FLD replaces Gaussian noise with ​​α-stable Lévy noise​​, enabling long jumps that facilitate escape from local minima. Theoretically, FLD achieves a ​​polynomial escape time​​ from local optima, outperforming LD's exponential escape time dependent on barrier height. FLD generalizes LD and adapts to diverse CO scenarios via α-tuning. The authors develop explicit-gradient (FLD-EG) and implicit-gradient/data-driven (FLD-IG) variants. Experiments on Maximum Independent Set, Maximum Clique, and Maximum Cut problems demonstrate that FLD advances both sampling-based and data-driven approaches, achieving ​​state-of-the-art performance​​ in most cases.

**Questions:**

See above.

**Ethical Concerns:**

["NO or VERY MINOR ethics concerns only"]

**Final Justification:**

I appreciate the responses in the author rebuttal. They are satisfacotry to me.

**Limitations:**

See above.

**Paper Formatting Concerns:**

No.

**Quality:**

3

**Strengths And Weaknesses:**

Strengths:

+ The paper provides rigorous analysis showing FLD achieves polynomial escape times (vs. LD's exponential dependence on barrier height) and generalizes LD (recovering it when α=2).

+ The work develops both explicit-gradient (FLD-EG) and implicit-gradient/data-driven (FLD-IG) variants, broadening applicability to problems with/without closed-form gradients.

Weaknesses:

- The energy functions of the proposed method use simple penalty terms (Eq. 6); complex constraints (e.g., multi-objective or conditional) remain unaddressed.

- The evaluations omit comparisons with evolutionary strategies (e.g., CMA-ES) or modern heuristics tailored to specific problems (e.g., MaxCut solvers).

- FLD-IG’s training efficiency is noted, but inference scalability for massive graphs (e.g., >10k nodes) needs deeper analysis.

- The proposed method has good theoretic results. However, the experimental results do not show significant advantage over existing methods.

---

> ### Author Rebuttal · Authors · 2025-07-30
>
> We sincerely appreciate the effort you put into reviewing our paper. Below, we provide our responses to the weaknesses and questions you raised.
>
> > **W1: The energy functions of the proposed method use simple penalty terms (Eq. 6); complex constraints (e.g., multi-objective or conditional) remain unaddressed.**
>
> Thanks for your constructive comment. While our paper employs a simple penalty term in the energy function and does not yet incorporate more sophisticated constraints, the scenarios we address are nonetheless highly challenging, and the case studies we present involve standard problems commonly used in prior work—on which we have achieved strong results. We believe that our proposed method can be extended to tackle more complex problems, and in future work we will expand our approach to include the elaborate penalty terms and complex constraints you mentioned.
>
> > **W2: The evaluations omit comparisons with evolutionary strategies (e.g., CMA-ES) or modern heuristics tailored to specific problems (e.g., MaxCut solvers).**
>
> Thank you for raising this question. As far as we know, CMA‑ES is typically applied to continuous optimization problems, whereas our work focuses on discrete combinatorial optimization. If you could kindly point us to any relevant literature, we would be most grateful. Furthermore, our KaMIS baseline employs the ReduMIS algorithm, which conducts an evolutionary search on a reduced graph. As for MaxCut solvers, we present the results of a semi-definite programming method (SDP), which is a widely used baseline for MaxCut, in the table below and will include them in the final version of our paper.
>
> | Method | Problem              | Size    | Time   |
> | ------ | -------------------- | ------- | ------ |
> | SDP    | MaxCut-BA-[200–300]  | 700.36  | 35.78m |
> | SDP    | MaxCut-BA-[800–1200] | 2786.00 | 10.00h |
>
> > **W3: FLD-IG’s inference scalability for massive graphs (e.g., >10k nodes) needs deeper analysis.**
>
> We apologize for any misunderstanding. In our ER‑dataset experiments, training was conducted on ER‑[700–800], while inference was performed on ER‑[9000–11000]. The results suggest that our method may exhibit reasonable inference scalability even on massive graphs.
>
> > **W4: The experimental results do not show significant advantage over existing methods.**
>
> Thank you for your question. We fully acknowledge your point: although our theoretical contributions are strong, the empirical improvements may not appear dramatic. Since prior solvers already approach the optimal solution, we focused on improving performance in the near‑optimal regime—which is substantially more challenging than improving cases far from optimal. Although our method yields improved results over previous approaches on most datasets, these gains are moderate and indicate that there is still plenty of scope for further refinement.
>
> We sincerely hope that our response provides you with a clearer and more thorough understanding of our work. If you have any additional questions, please don't hesitate to reach out. We would be more than happy to address any concerns.

---

> > ### Comment · Reviewer_ymVV · 2025-08-04
> > **The responses are satisfacotry.**
> >
> > I appreciate the responses in the author rebuttal. They are satisfacotry to me.

---

> > > ### Author Response · Authors · 2025-08-04
> > >
> > > Thank you for taking the time to review our rebuttal. We’re glad the clarifications addressed your concerns. Please let us know if any additional information would be helpful.

---

### Official Review · Reviewer_YtYc · 2025-07-02

**Clarity:** 3
**Significance:** 3
**Originality:** 3
**Rating:** 5
**Confidence:** 2

**Summary:**

This paper addresses a known limitation of the application of Langevin dynamics (LD) for Combinatorial Optimization (CO) : its poor ability to escape local optima, especially in rugged landscapes. To that end, they propose replacing the Gaussian noise with \alpha-stable Lévy noise, introducing the Fractional Langevin Dynamics framework (FLD). This approach allows occasional large steps (Lévy flights) to escape local minima and provide a better coverage of the solution space. Authors compare the upper bounds of mean escape time and show that this quantity scales in polynomial time with FLD, an improvement over the exponential bound provided by LD.  Authors then show how to derive from FLD a sampling method, where stability is dealt with using a truncation scheme and a data-driven method based on [Feng and Yang, 2025]. Performance of the method is evaluated on 3 common CO problems, showing SOTA results in most applications. Impact of the \alpha parameter is also investigated.

**Questions:**

See weaknesses

**Ethical Concerns:**

["NO or VERY MINOR ethics concerns only"]

**Limitations:**

Authors have adequately addressed the limitations of their work, claims are theoretically justified and experiments are fair

**Quality:**

3

**Strengths And Weaknesses:**

Strengths:
    - The paper is mostly clear and well written
    - Replacing Gaussian noise with \alpha-stable Lévy noise is sound and is well justified
    - Claims are theoreticvally and empirically justified, showing an clear improvement over SOTA
    - Experiments are clean with a high number of baselines and it is appreciated that running time are provided.

Weaknesses:
    - Authors do not dive deep into the truncation strategy introduced to improve stability for low \alpha. Does it potentially introduce a bias? Is it difficult to tune?
    - Could authors add variance accross runs to ensure consistency?
    - Is the idea of replacing the Gaussian noise with \alpha-stable Lévy noise in LD novel? Has it been already done in adjacent fields? More generally, variants of LD could have been explored in the related work.
    - Section 3.5 is unclear to me. Could the authors provide more details in the main paper to clarify it ? Especially regarding the PRL procedure

---

> ### Author Rebuttal · Authors · 2025-07-30
>
> We sincerely appreciate the effort you put into reviewing our paper. Below, we provide our responses to the weaknesses and questions you raised.
>
> > **W1: Authors do not dive deep into the truncation strategy introduced to improve stability for low $\alpha$.**
>
> We sincerely apologize for the lack of clarity in our description, which may have caused confusion. Since we initially applied our method to 0-1 integer programming problems, for problems where the values are limited to 0 and 1, large outliers can cause the solution to remain stuck at 0 or 1, which goes against our intention of using noise to facilitate escaping from local minima. Therefore, applying reasonable truncation to the outliers allows the iteration process to more easily escape from local optima. We also show in the table below that $d_{\mathrm{noise}}$ is easy to tune, and slight perturbations to $d_{\mathrm{noise}}$ do not significantly affect the performance.
>
> | $d_{\mathrm{noise}}$ | Problem             | Size   |
> | --------- | ------------------- | ------ |
> | 20        | MIS-ER-[700–800]    | 44.37  |
> | 30        | MIS-ER-[700–800]    | 44.34  |
> | 40        | MIS-ER-[700–800]    | 44.35  |
> | 2         | MaxCl-RB-[200–300]  | 18.96  |
> | 4         | MaxCl-RB-[200–300]  | 18.97  |
> | 6         | MaxCl-RB-[200–300]  | 18.97  |
> | 20        | MaxCut-BA-[200–300] | 734.06 |
> | 30        | MaxCut-BA-[200–300] | 734.13 |
> | 40        | MaxCut-BA-[200–300] | 734.16 |
>
> > **W2: Could authors add variance accross runs to ensure consistency?**
>
> We sincerely apologize for not including the variance of the datasets in the paper to ensure consistency. We evaluate the variance of FLD-EG by using different seeds for 10 repeated experiments, and the results are reported below:
>
> | Problem              | Size         |
> | -------------------- | ------------ |
> | MIS-RB-[200–300]     | 20.02±0.03   |
> | MIS-RB-[800–1200]    | 40.24±0.04   |
> | MIS-ER-[700–800]     | 44.34±0.09   |
> | MIS-ER-[9000–11000]  | 377.28±1.07  |
> | MaxCl-RB-[200–300]   | 18.97±0.01   |
> | MaxCl-RB-[800–1200]  | 40.61±0.06   |
> | MaxCut-BA-[200–300]  | 734.09±0.25  |
> | MaxCut-BA-[800–1200] | 2960.06±1.68 |
>
> We will include the variance for all results of our methods in the final version of our paper.
>
> > **W3: Is the idea novel? Has it been already done in adjacent fields?**
>
> Thank you for your question. We sincerely acknowledge that replacing Gaussian noise with $\alpha$-stable L\'evy noise is not an entirely new strategy. However, it has not been explored in the field of CO problems. We recognized the potential of $\alpha$-stable L\'evy noise in CO problems and have made preliminary attempts. Moving forward, we plan to extend these experiments to more complex problems. We apologize for not providing a detailed discussion of the application of FLD in other fields due to space limitations. We will include a more comprehensive review of the related work in the final version of our paper.
>
> > **W4: Section 3.5 is unclear to me. Could the authors provide more details in the main paper to clarify it ?**
>
> We sincerely apologize for not providing a detailed description of FLD-IG in the main paper. Due to space limit, we included the specific method details and pseudocode **in Appendix E.2**. However, we would like to provide a more detailed explanation of the FLD-IG methods here:
>
> Based on the differentiability of the energy function, we divide the FLD-based approach to solving CO problems into explicit gradient (FLD-EG) method in Section 3.4 and implicit gradient (FLD-IG) method in Section 3.5. In Section 3.5, when the energy function is not differentiable, we introduce an NN-based framework whose training procedure resembles reinforcement learning, alternating between sampling and update steps, and using auto-grad to approximate the gradient of the energy function.
>
> We sincerely hope that our response provides you with a clearer and more thorough understanding of our work. If you have any additional questions, please don't hesitate to reach out. We would be more than happy to address any concerns.

---

### Official Review · Reviewer_MK47 · 2025-07-02

**Clarity:** 2
**Significance:** 3
**Originality:** 3
**Rating:** 4
**Confidence:** 3

**Summary:**

The paper proposes a novel sampling technique based on fractional Langevin dynamics and incorporates it into a method for solving combinatorial optimization problems with binary variables and equality constraints. The key feature of this new sampling strategy is its ability to escape narrow local minima more quickly and better handle the challenging landscape of the objective function. The main math tool to develop such sampling is $\alpha$-stable Levy noise, which replaces Gaussian noise in the classical Langevin dynamics. The authors theoretically show the polynomial time for escaping local minima, which is the solid basis for the presented approach. In experiments, the study demonstrates a better trade-off between the maximum values of objective functions and runtime compared to many competitors. The authors select MaxCut, MaxClique, and Maximum Independent Set problems as the benchmarks for testing the presented approach.

**Questions:**

1. Why Eq. (6) uses $+ \lambda b(x)$. and not $ + \lambda b^2(x)$ ? A large negative $b(x)$ leads to a small value of $H$, but it violates the constraint in the initial problem.
2. In line 207, the authors first mention 'gradient' in the context of sampling and do not provide any clarification of what this term means. Please add necessary remarks about what gradient is used and why it is "explicit."? Is it possible to use "implicit gradient"? I find this term in the next section; however, its meaning remains unclear to me.
3. What is the effect of the truncation parameters $d, d_{noise}$ and $\lambda$ in Eq. (29)?
4. What $\alpha$ is used in the ablation study for FLD?
5. What is $d$ in Eq. (29)?
6. Could the authors please perform an additional experiment on the toy problem to illustrate the feature of the fractional Langevin dynamics to escape from the narrow local minima faster than the classical Langevin dynamics? I think such a simple benchmark will not require too much space, but will serve as a bridge between advanced theory and challenging benchmark problems.

**Ethical Concerns:**

["NO or VERY MINOR ethics concerns only"]

**Final Justification:**

The paper proposes a motivated modification to the noise distribution used in the sampling process for solving the combinatorial optimization problem. The study includes both solid theoretical analysis and convincing experimental evaluation of the suggested approach. The authors address my concerns in rebuttal, so I maintain my score and lean towards acceptance of this work.

**Limitations:**

A limitation on the binary variables is mentioned in the conclusion section.

**Paper Formatting Concerns:**

1. The sentence in lines 190-193 is too long; please split it to improve readability.

**Quality:**

3

**Strengths And Weaknesses:**

Strenghts
1. The paper is mostly well-written, and the authors' motivation for developing a better sampling strategy is clear.
2. The experimental evaluation looks convincing and promising since the proposed approach outperforms many learning-based competitors and the standard OR tool Gurobi.
3. The theoretical introduction clearly explains the basic concepts and describes key advanced concepts, such as the symmetric $\alpha$-stable distribution.

Weaknesses
1. The description of the FLD-IG and FLD-EG methods is too brief. Since they build upon the central theoretical framework presented in the paper, a clearer exposition of these methods would strengthen the paper's impact.
2. I do not find the explicit statement about the overall pipeline complexity and memory consumption. Is the total runtime smaller due to the smaller number of iterations or the lower cost of a single iteration?
3. The authors compare the developed schemes with learning-based methods; however, it is unclear to me whether the proposed methods have a training stage. If yes, then how long is this stage? Is it included in the runtime comparison in Tables 1 and 2?

---

> ### Author Rebuttal · Authors · 2025-07-30
>
> We sincerely appreciate the effort you put into reviewing our paper. Below, we provide our responses to the weaknesses and questions you raised.
>
> > **W1: The description of the FLD-IG and FLD-EG methods is too brief.**
>
> We sincerely apologize for not providing a detailed description of FLD-IG and FLD-EG in the main paper. Due to space limitation, we included the specific method details and pseudocode **in Appendix E**. However, we would like to provide a more detailed explanation of the FLD-IG and FLD-EG methods here:
>
> Based on the differentiability of the energy function, we divide the FLD-based approach to solving CO problems into explicit gradient (FLD-EG) and implicit gradient (FLD-IG) methods. For FLD-EG, we provide a recursive expression with gradient and noise truncation, where a multi-step sampling and recursion approach can approximate the optimal solution. For FLD-IG, when the energy function is not differentiable, we introduce an NN-based framework whose training procedure resembles reinforcement learning, alternating between sampling and update steps, and using auto-grad to approximate the gradient of the energy function.
>
> We will include this detailed explanation of the FLD-IG and FLD-EG methods in the final version of our paper.
>
> > **W2: Not find the explicit statement about the overall pipeline complexity and memory consumption.**
>
> We sincerely apologize for not providing the complexity and memory consumption of the method pipeline in the main paper. Compared with the RLSA and RLNN methods, the complexity in the single-step sampling phase of FLD-IG and FLD-EG is similar. However, due to the stronger escape ability of FLD with symmetric $\alpha$-stable noise compared to LD with Gaussian noise, the FLD-IG and FLD-EG methods have fewer iteration steps. In contrast to other learning-based methods, FLD-IG and FLD-EG only involve sampling and updating steps, resulting in a lower cost in a single iteration.
>
> The memory consumption is summarized in the table below:
>
> |        | MIS-RB-[200–300] | MaxCl-RB-[200–300] | MaxCut-BA-[200-300] |
> | ------ | ---------------- | ------------------ | ------------------- |
> | FLD-EG | 1576.96MB        | 1413.12MB          | 1372.16MB           |
> | FLD-IG | 1761.28MB        | 1536.00MB          | 1556.48MB           |
>
> Here, we report only a single result for each case; in the final version of our paper, we will include the complete set of experimental results.
>
> > **W3: Does the proposed methods have a training stage?**
>
> The FLD-EG method is a training-free method, while the FLD-IG method involves a training stage. The time consumption is shown in the table below. In Tables 1 and 2, the runtime includes only the model inference time and does not account for the time spent in the training stage. However, the training stage does not take very long time, and for each problem, the training stage will be carried out only once.
>
> |        | MIS-RB-[200–300] | MaxCl-RB-[200–300] | MaxCut-BA-[200-300] |
> | ------ | ---------------- | ------------------ | ------------------- |
> | FLD-IG | 4.65h            | 7.83h              | 4.72h               |
>
> Here, we report only a single result for each case; in the final version of our paper, we will include the complete set of experimental results.
>
> **To Q1**: We sincerely apologize for any confusion caused by the phrasing in our paper. Equation (6) represents the Lagrangian function, not a pure penalty function. This linear term naturally arises from the dualized constraints in dual theory, ensuring that when the KKT conditions are satisfied, the primal and dual problems are equivalent (strong duality holds for convex problems).
>
> **To Q2**: In Equation (20), during the discrete sampling process, there is the gradient of the energy function. If the energy function is differentiable, the explicit gradient can be directly incorporated into the recursive expression, and the optimal solution is approximated through iterative updates in the sampling steps. This is referred to as FLD-EG. If the energy function is non-differentiable, the gradient of the energy function is implicitly approximated using auto-grad, which is referred to as FLD-IG.
>
> **To Q3&Q5**:  The truncation parameter $d$ is applied for both the gradient and the $\mathcal{S}\alpha \mathcal{S}$ noise in the Equation (29). We apply the Top-$d$ truncated indicator to $\nabla H$, ensuring that only the $d$ components with the largest magnitude influence the update and apply the Top-$d_{\mathrm{noise}}$ truncated indicator to the sampled noise vector, truncating extreme values to stabilize the sampling process. The second term in the Equation (29) regularizes the expected Hamming distance between the two solutions, with $\lambda'$ being the regularization coefficient.
>
> **To Q4**: The value of $\alpha$ in the ablation study is consistent with the one presented in Table 3 and Table 4.
>
> **To Q6**: We sincerely appreciate your valuable suggestions. In the ablation study, we explored the use of "fractional Langevin dynamics to escape from the narrow local minima faster than the classical Langevin dynamics." Below, we present specific data to demonstrate the escape ability of FLD.
>
> As external links aren’t permitted in the rebuttal phase, we’re unable to include images here. Instead, we provide a textual description below to address your question: For the MIS problem on an ER‑[700–800] instance, LD required 50 steps to reach the Best Energy of –39, whereas FLD did so in just 16 steps. To attain the Best Energy of –41, LD needed 210 steps, compared to only 100 steps for FLD. Likewise, on a BA‑[200–300] instance of the MaxCut problem, LD took 32 steps to reach –800, while FLD achieved the same in just 8 steps.
>
> **To Paper Formatting Concerns**: Thanks for your kind suggestion. We have rewrite the sentence in lines 190-193: "Similarly, we provide the discrete proposal for the escape time of both LD and FLD. We state upfront that the Markov chains of LD and FLD are reversible if they satisfy the detailed balance conditions. Additionally, $p_{\tau}(x)$ is a positive stationary distribution, given that the symmetric proposal and the Metropolis-Hastings acceptance criterion are satisfied for constructing discrete LD and FLD."
>
> We sincerely hope that our response provides you with a clearer and more thorough understanding of our work. If you have any additional questions, please don't hesitate to reach out. We would be more than happy to address any concerns.

---

> > ### Comment · Reviewer_MK47 · 2025-08-04
> >
> > Dear authors,
> >
> > Thank you for the detailed response! No more questions from my side. I believe that, after minor revisions in response to the reviewers' comments, the paper makes a significant contribution, so I maintain my score as it is.

---

> > > ### Author Response · Authors · 2025-08-04
> > >
> > > We appreciate your careful review of our rebuttal and are encouraged that the clarifications were satisfactory. We remain available to supply any further information.

---

### Note · Authors · 2025-08-12

We sincerely thank the reviewers for their time and constructive feedback.
 Throughout the rebuttal process, we have carefully addressed all questions and concerns raised by the reviewers:

1. **Clarifications and Missing Details:** We have expanded explanations of the FLD-IG and FLD-EG methods, provided precise definitions of parameters, and clarified the role of the truncation strategy, gradient types, and training stage. Additional tables were included to report runtime, memory consumption, and variance across multiple seeds.
2. **Experimental Completeness:**  We added comparative results with additional baselines (e.g., SDP for MaxCut), discussed CMA-ES applicability, and provided more information on inference scalability for large graphs.
3. **Theoretical and Empirical Balance:** We clarified that Equation (6) is derived from a Lagrangian formulation rather than a penalty term, ensuring theoretical soundness. We also elaborated on potential extensions to more complex constraints and objectives in future work.
4. **Formatting and Readability:**  We addressed minor presentation issues, improved equation notation, and committed to expanding the Related Work section in the final version.

Finally, we believe these clarifications and additional results have resolved the reviewers’ uncertainties. The overall reviewer ratings and comments acknowledge the novelty, solid theoretical foundation, and competitive empirical performance of our method. We sincerely thank all reviewers and the AC for their valuable time and effort.

---

### Decision · Program_Chairs · 2025-09-17

**Decision:**

Accept (poster)

**Comment:**

This paper proposes Fractional Langevin Dynamics for the combinatory optimization problem. All reviewers acknowledge the technical contributions of this paper and are also satisfied with the empirical demonstration. The authors provide sufficient feedbacks during the rebuttal phase and the concerns of some reviewers are well addressed. Therefore, I would lean to accept this paper.